# Intense pH Sensitivity Modulation in Carbon Nanotube-Based Field-Effect Transistor by Non-Covalent Polyfluorene Functionalization

**DOI:** 10.3390/nano13071157

**Published:** 2023-03-24

**Authors:** Gookbin Cho, Eva Grinenval, Jean-Christophe P. Gabriel, Bérengère Lebental

**Affiliations:** 1Laboratoire de Physique des Interfaces et des Couches Minces, LPICM, CNRS, Ecole Polytechnique, Institut Polytechnique Paris, 91128 Palaiseau, France; 2LICSEN, NIMBE (UMR CEA/CNRS 3685), Université Paris-Saclay, 91191 Gif sur Yvette, France; 3IMSE, COSYS, Université Gustave Eiffel, Marne-la-Vallée Campus, 77447 Marne-La-Vallée, France

**Keywords:** carbon nanotubes, field effect transistor, electrolyte gating, pH sensing, non covalent functionalization, polyfluorene, urea, pH buffer

## Abstract

We compare the pH sensing performance of non-functionalized carbon nanotubes (CNT) field-effect transistors (p-CNTFET) and CNTFET functionalized with a conjugated polyfluorene polymer (labeled FF-UR) bearing urea-based moieties (f-CNTFET). The devices are electrolyte-gated, PMMA-passivated, 5 µm-channel FETs with unsorted, inkjet-printed single-walled CNT. In phosphate (PBS) and borate (BBS) buffer solutions, the p-CNTFETs exhibit a p-type operation while f-CNTFETs exhibit p-type behavior in BBS and ambipolarity in PBS. The sensitivity to pH is evaluated by measuring the drain current at a gate and drain voltage of −0.8 V. In PBS, p-CNTFETs show a linear, reversible pH response between pH 3 and pH 9 with a sensitivity of 26 ± 2.2%/pH unit; while f-CNTFETs have a much stronger, reversible pH response (373%/pH unit), but only over the range of pH 7 to pH 9. In BBS, both p-CNTFET and f-CNTFET show a linear pH response between pH 5 and 9, with sensitivities of 56%/pH and 96%/pH, respectively. Analysis of the I–V curves as a function of pH suggests that the increased pH sensitivity of f-CNTFET is consistent with interactions of FF-UR with phosphate ions in PBS and boric acid in BBS, with the ratio and charge of the complexed species depending on pH. The complexation affects the efficiency of electrolyte gating and the surface charge around the CNT, both of which modify the I–V response of the CNTFET, leading to the observed current sensitivity as a function of pH. The performances of p-CNTFET in PBS are comparable to the best results in the literature, while the performances of the f-CNTFET far exceed the current state-of-the-art by a factor of four in BBS and more than 10 over a limited range of pH in BBS. This is the first time that a functionalization other than carboxylate moieties has significantly improved the state-of-the-art of pH sensing with CNTFET or CNT chemistors. On the other hand, this study also highlights the challenge of transferring this performance to a real water matrix, where many different species may compete for interactions with FF-UR.

## 1. Introduction

In recent years, rapid population growth and unsustainable water use in agriculture and industry have led to global hydric stress [1]. To mitigate the global drinking water crisis, water quality monitoring is of great interest. To reduce its cost and improve its performances, many researchers have proposed nanomaterial-based water quality sensors [2,3,4]. Among the various nanomaterial options, carbon nanotube (CNT) sensors [5] have been intensively studied for chemical sensing in water due to their excellent mechanical and chemical stability, their large surface area and their chemical tunability, which enables selective sensing [6,7,8].

One of the most important parameters to monitor is pH (concentration of H+ ions), since its variations can indicate harmful events (e.g., bacterial contamination, ingress of contaminated water) and knowledge of this allows the optimization of water treatment (e.g., chlorination). CNT devices have been tested as sensors for numerous parameters relevant to water quality, and pH is the most studied among them [9], either as the main target or as a key interferent for the measurement of another quantity.

While there has been a longstanding interest in CNT-based optical sensors [10], their long-term applications are usually targeted at ex-situ (laboratory) water testing or point-of-care diagnostics. Electrochemical CNT sensors have also been intensively studied [9], as this mode of transduction (with or without CNT) is already widely used in commercial pH sensors. Their use cases include both ex-situ and in-situ water monitoring. However, while they can achieve excellent accuracy, they have serious shortcomings, in particular the limited stability of the electrodes in use and in storage, so that the sensors require frequent recalibration and/or replacement of consumables. The cost of commercial products is still too high for widespread use.

While electrochemical pH sensors rely on monitoring the exchange of electrons from one electrode to another through the target solution, in electrical pH sensors (either chemistors or field-effect-transistors (FET) [11]), the pH of the target solution perturbs the intrinsic electrical properties of a metallic or semiconducting layer, and this perturbation is detected by monitoring the electrical behavior of the device. As solid-state devices (with no need for electrolyte storage), such electrical sensors are promising for future in-situ water monitoring solutions: compared to commercial pH sensors, they are expected to be cheaper to fabricate and easier to store, to require less calibration, to be more durable under field conditions, and to be less dependent upon the composition of the target water matrix. This led to the first demonstration and proposed application of their pH sensitivity for the fabrication of a capnometer to track the level of CO_2_ in the breath, whose electrical modulation was induced by the change of pH at the surface of the CNTs. The latter effect was provoked by the reversible dissolution of CO_2_ and the associated formation of carbonic acid in the water-rich functionalization layer [12,13].

Table 1, expanded from [9,14], reviews the literature on electrical CNT-based pH sensors, i.e., chemistors and field-effect transistors. The detection limit of pH is rarely reported in the literature, so the relative sensitivity was calculated here as a means to compare between the different transduction modes and operating protocols.

CNT-based chemistors consist of a layer of CNTs (either SWCNTs or MWCNTs) positioned between two metallic electrodes [8]. The devices are characterized by different types of electrode material, substrate, CNT deposition mode and, most importantly, CNT functionalization strategy. While the reported devices have performances ranging from 3.5%/pH unit to 18%/pH unit, with an average of 11%/pH unit in the literature, the best performance so far has been achieved with non-functionalized CNTs [15]. In this case, the pH sensitivity is conferred by the intrinsic carboxylic functional groups on the CNT sidewalls.

In comparison, CNTFETs also consist of a CNT layer between two electrodes, but in this case, the CNT layer is semiconducting [16,17]. This is usually achieved by using SWCNTs that are either deposited at low concentrations with or without pre-sorting [7,18] or are grown in place at low concentrations with or without chirality control, allowing wafer-scale fabrication [19]. A third (and sometimes fourth) electrode is then used to modulate the electric-field within the CNT, enabling field-effect-based modulation. Most CNTFETs used as pH sensors are liquid-gated [20], as this approach has the simplest fabrication route. It also provides excellent transistor performances because the electrical double layer formed in the electrolyte acts as a dielectric layer [21]. In practice, the gate voltage is applied directly through an independent, commercial electrode or through the device itself via a side electrode or an exposed bottom gate. Alternatively, top gating or dual (top and bottom) gating is possible, although it is more difficult to fabricate because the top dielectric and top gate require additional processing steps and they both must be porous to expose the CNT layer to the target solution. CNTFET performance ranges from 8%/pH unit to 24%/pH unit, with an average of 12%/pH unit. While this average relative sensitivity is very close to that of chemistors, the best performance (24%/pH unit) is significantly higher than what has been achieved with chemistors so far. It is again achieved with non-functionalized CNT, in a dual-gated FET configuration [22].

In the present work, we demonstrate state-of-the-art pH sensing performances (25%/pH unit in phosphate buffer) with inkjet-printed, effectively-liquid-gated CNTFETs using unsorted, non-functionalized SWCNTs. In addition, we show that the functionalization of these SWCNTs with a polyfluorene polymer bearing urea moieties (labeled FF-UR) results in an even greater improvement in performance (373%/pH unit—from pH 7 to 9 only—in phosphate buffer). After detailing the fabrication and characterization process in air and water (the latter being optimized to ensure stability of measurements), we compared the pH sensing performances in different pH buffer solutions and over time. We then discuss the sensing mechanisms, in particular how pH sensitivity modulation is achieved with FF-UR.

**Table 1 nanomaterials-13-01157-t001:** CNT-based pH sensors in water, sorted by transduction type and then by relative sensitivity. Average relative sensitivities for chemistors and FETs are 11%/pH unit and 12%/pH unit, respectively. Expanded from [9].

Type of CNT	Functional Probe	Functionalization	Detection Range	Sensitivity	Relative Sensitivity *	Transduction Method	CNT Deposition Method	Electrode Material Contact Configuration	Substrate	Ref.
SWCNT	Polyaniline	Non covalent	pH 2.1–12.8	N/A	N/A	Chemistor	Drop-casting	Ti/Au	Si/SiO_2_	[23]
SWCNT	Nafion	Non covalent	pH 1–12	N/A	3.5%/pH	Chemistor	Screen printing	SWCNT	Polymide	[24]
MWCNT	Ni NP *	Non covalent	pH 2–10	N/A	5.0%/pH	Chemistor	Continuous pulling of super-aligned, CVD grown MWCNTs	MWCNT	PDMS	[25]
SWCNT	Pristine	Non functionalized	pH 1–11	34 nS/pH (pH 1–6) 163 nS/pH (pH 7–11)	3.4%/pH (pH 1–6) 9.3%/pH (pH 7–11)	Chemistor	Spray-casting	Cr	Si/SiO_2_	[26]
SWCNT	COOH	Covalent	pH 5–9	75 Ω/pH	11%/pH	Chemistor	Dielectrophoresis (aligned CNTs)	Cr/Au	Si/SiO_2_	[27]
SWCNT	Pristine	Non functionalized	pH 4–10	5.2 kΩ/pH	14%/pH	Chemistor	Aerosol jet printing	Ag	Kapton	[28]
SWCNT	Polyaniline/ PVA	Non Covalent	pH 1–10	15 kΩ/pH	~15%/pH	Chemistor	Layer by layer assembly	None	Glass	[29]
MWCNT	Pristine	Non functionalized	pH 5–9	63 Ω/pH	18%/pH	Chemistor	Sucked by vacuum force	MWCNT	Filter paper	[15]
SWCNT	ETH500 *, MDDA-Cl	Non covalent	pH 2–7.5	71 nA/pH	8%/pH	FET Side-gated Effectively liquid-gated	Spray deposition	Aqueous electrolyte (gate) Cr/Au (5/50 nm)	Polymide (Kapton^®^)	[30]
SWCNT	PDDA	Non Covalent	pH 5–9	~23 µA/pH	~8%/pH	FET liquid-gated	Layer by layer assembly	Cr/Au (25 nm/100 nm)	PET	[31]
SWCNT	COOH	Covalent	pH 3–8	17 nA/pH	8%/pH	FET top-gated	N.P.	Cr/Au (30/50 nm) source & drain electrodes, Ag/AgCl for reference electrode	Glass/APS(50–200 nm)/SWCNT /APS(500 nm)/TopGate	[32]
SWCNT	Pristine	Non functionalized	pH 3.4–7.8	3.9 µA/pH	13%/pH	FET Bottom-gated Effectively liquid-gated	Spin coating	Cr/Au (5/40 nm)	Si/SiO2(65 nm)	[33]
SWCNT	Poly(1-aminoanthracene)	Non covalent	pH 3–11	FET 19 µS/pH potentiometry 55 mV/pH	FET 14%/pH potentiometry N/A	FET (liquid gated)	Dielectrophoresis (aligned CNTs)	Au contacts, Pt wire (Auxillary), Ag/AgCl electrode (Reference)	Si/SiO_2_(300 nm)	[34]
SWCNT	Pristine	Non functionalized	pH 3–10	7600 mV/pH (Dual-gate mode) 59.5 mV/pH (single-gate mode potentiometry)	23%/pH (Dual-gate mode) N/A (single-gate mode potentiometry)	FET Double-gated (bottom and top)	Spin coating	100 nm Ti contacts for source, drain and top gate	p-Si (substrate acting as bottom gate)	[22]
SWCNT	Pristine	Non functionalized	pH 3–9 (PBS) pH 5–9 (BBS)	91.7 nA/pH (PBS) 0.37 µA/pH (BBS)	25%/pH (PBS) 56%/pH (BBS)	FET Bottom-gate Effectively liquid gated	Ink-jet printing	Ti/Pt (50/200 nm)	Si/SiO_2_ (1000 nm)	This paper
SWCNT	FF-UR polyfluorene polymer carrying urea moieties	Non covalent	pH 7–9 (PBS) pH 5–9 (BBS)	2.8 nA/pH (pH 3–6 PBS) 65.1 nA/pH (pH 7–9 PBS) 0.21 µA/pH (BBS)	16%/pH (pH 3–6 PBS) 373%/pH (pH 7–9 PBS) 96%/pH (BBS)	FET Bottom-gate Effectively liquid gated	Ink-jet printing	Ti/Pt (50/200 nm)	Si/SiO_2_ (1000 nm)	This paper

* The relative sensitivity is calculated using the formula Relative Sensitivity = (x/x_0_) ∗ 100(%), with x the absolute sensitivity expressed (depending on the transduction) in units of resistance, voltage or current per pH unit and x_0_ the baseline parameter (resistance, voltage or current) at pH 7. The relative sensitivity is not calculated for potentiometry and voltammetry-based transduction as it depends on the choice of reference voltages. Acronyms: N.P.: not provided; Ni NP: Nickel nanoparticle; PDMS: Polydimethylsiloxane; PDDA: poly(diallyldimethyammonium); MDDA-Cl: methyltridodecylammonium chloride; ETH500: tetradodecylammonium tetrakis(4-chlorophenyl)borate.

## 2. Materials and Methods

### 2.1. CNTFET Substrate

A bottom gated CNTFET structure was initially selected (Figure 1) because it allows the CNTFET to be tested in air and provides maximum CNT exposure to water. If all wirebond contacts are fully passivated (here, using waterproof globtop), the system can operate as a bottom-gated FET in water; if not, the CNTFET is effectively liquid-gated, with the liquid gate voltage being applied through the wirebond contact pads of the bottom gate.

The substrate was a 4-inch undoped silicon wafer with a thickness of 525 μm and a resistivity of up to 20 Ω∙cm. The wafer was first thermally oxidized to obtain a 1.0 µm of SiO_2_ layer as insulator. Then, a 400 nm-thick poly-Si layer was deposited, and doped by Boron ion implantation at 40 keV to achieve a doping concentration of 5∙10^15^ at/cm^2^. After another heating process at 1050 °C for 30 min, the doping concentration was increased to 7∙10^19^ at/cm^2^. Then, two different insulating layers were deposited—a 200-nm-thick SiO_2_ layer and a 20-nm-thick Si_3_N_4_ layer—to avoid any interface problem between water and SiO_2_ [35]. The top metal layer (source and drain electrodes and contact pads) consisted of a 200 nm Pt layer with a 50 nm Ti adhesion layer. Pt was chosen instead of Au because gold can react with sodium or potassium cyanide under alkaline conditions in the presence of oxygen to form soluble complexes [36].

The chip and electrode designs are shown in Figure 2. The electrodes were patterned by photolithography (two mask layers). Each unit chip was 10 mm by 10 mm dimensions and contained 32 CNTFET devices (four transistors each for eight different channel lengths). There were 45 chips per 4” wafer. The source and drain electrodes measured 1100 µm by 250 µm. The distance between the source and drain electrodes defines the channel length and varied from 5 µm to 120 µm. The gate electrodes were 1250 µm long. The gate width was equal to the channel length. The chips were fabricated by the Ecole Supérieure d’Ingénieurs en Electrotechnique et Electronique de Paris (ESIEE-Paris), according to our design.

### 2.2. CNT Ink

The CNT-ink preparation process was adapted from our previous work [37]: 1 mg of unsorted single-walled carbon nanotube (HiPCo SWCNT, 95+% purity, 70% semiconducting, from Nanointegris^TM^, Boisbriand, QC, Canada) was added to 100 mL of 1-methyl-2-pyrrolidinone (NMP, 99+% purity, Sigma Aldrich), corresponding to 0.001 wt%. This SWCNT/NMP mixture was sonicated with a high-power tip sonicator (Vibra-cell^TM^ ultrasonic liquid processor, Sonics^TM^, Newtown, CT, USA) at 30 W for 1 h. The remaining bundles were then separated by centrifugation for 2 h at 10,000 RCF, and only the supernatant was used as printable ink.

### 2.3. CNT Functionalization

For CNT functionalization, we used a conjugated polymer (Figure 3) patented by our team for sensing applications [38]. The polymer, here called FF-UR, is composed of a fluorene backbone. The fluorene moieties are functionalized with either two alkyle chains to increase solubility and interaction strength with CNTs, or with two identical sensing moieties, a urea group NH-CO-NH between two phenyl groups (forming diphenylurea). The ability of this polymer to non-covalently functionalize CNTs has been demonstrated using molecular dynamics by Benda et al. [39], while its sensing capabilities have been analyzed using density functional theory with an implicit solvent model in [40]. Briefly, the urea group is expected to complex anions through H-bonds (notably glyphosate, hypochlorous ions), while for cations, cation-pi interactions with the phenyl groups are enhanced through interaction with the oxygen of the urea. This polymer is stable in the pH range of 3 to 11.

For CNT functionalization, FF-UR was first dissolved in NMP by magnetic stirring at room temperature for 12 h at a concentration of 1.5 mg/100 mL, which corresponds to the targeted mass ratio between CNT and polymer of 1:1.5. Finally, the CNT ink and the FF-UR solution were mixed together and then sonicated in a bath-type sonicator for 1 min at 25 °C for non-covalent functionalization.

In the remainder of this report, non-functionalized CNTs are referred to as p-CNTs, while FF-UR-functionalized CNTs are referred to as f-CNTs.

### 2.4. Ink-Jet Printing

As-prepared SWCNT ink was introduced into a cartridge (DMP-11601, Fujifilm^TM^, Santa Maria, CA, USA) and the ink was printed with a commercial high-resolution ink-jet printer (DMP-2800, Fujifilm^TM^, Santa Maria, California, USA). During the printing process, the temperature of a cartridge was set at 20 °C and the temperature of a substrate was set at 50 °C. Two layers were printed between the source and drain with a 300 μm by 300 μm square pattern (25 μm drop spacing) to create a semiconducting, percolating CNT network [41]. After inkjet printing, the CNT layer was annealed at 160 °C for 12 h to remove the remaining solvent. An example of the resulting deposition is shown in Appendix B, Figure A1.

For inkjet printing of functionalized CNT, the same procedure as for p-CNTFET was used. As-printed f-CNTFETs were annealed at 80 °C for 24 h to remove residual solvent. The annealing temperature for f-CNTFET was lower than that for p-CNTFET to avoid any damage to FF-UR by thermal degradation.

### 2.5. Passivation

The passivation step is mainly aimed at avoiding physical degradation of the device or detachment of the CNT in aqueous solution. It is also known to improve the operational performance of CNTFETs [42,43]. The passivation process was performed as follows: a PMMA/toluene (PMMA: molecular weight 15,000 from Acros Organics; toluene: anhydrous at 99.8% from Sigma Aldrich, Saint-Quentin-Fallavier, France) solution was prepared at a concentration of 5 mg/mL and spin-coated in two steps; 60 s at 1000 rpm and then 90 s at 3000 rpm. Since the PMMA layer needs to be porous for sensing application, a non-solvent induced phase separation process (NIPS) was then performed [44]. The chip was immersed for 1 min in a CaCl_2_ bath (5 wt%). This treatment induces a phase separation of the PMMA film into a polymer-rich phase and a polymer-poor phase, the latter forming the pores. After NIPS, the chips were annealed at 80 °C for 12 h.

### 2.6. Electrical Characterization in Air

To validate FET operation, CNTFETs were characterized in air under a probe station using a Keithley 4200A-SCS parameter analyzer (Tektronix^TM^, Beaverton, OR, USA). The applied drain-to-source voltage (V_ds_) was fixed at +5 V, and the gate-to-source voltage (V_gs_) was swept down from +60 V to –60 V in 0.1 V increments, then left at –60 V for 5 to 10 s, and then swept up from –60 V to +60 V with the same increment. The acquisition time of each measurement was set to 1 s. This wide gate voltage range is required to allow the modulation in air of this particular bottom-gated CNTFET architecture [43].

### 2.7. Electrical Characterization in Water

PMMA passivated chips were wire-bonded directly onto custom-designed printed circuit boards (PCBs) using a semi-automatic wire bonder in wedge mode (iBond 5000, Micro Point Pro^TM^, Yokne’am Illit, Israel) (Appendix A). To prevent tearing during handling in water, the wire bonds were protected with Norland UV resist (manually applied). However, it should be noted that this resist does not fully completely waterproof the wire bonds and associated pads. In fact, electrolysis of water [45] was visually observed along the contact pads when a voltage greater than 1 V was applied, (Appendix C, Figure A2). Therefore, it was expected that the CNTFET, although bottom-gated in air, would effectively operate as a liquid-gated FET in water, which was confirmed by the characterizations presented in the following sections.

For measurement in water, the PCB-mounted chips were measured with a Keithley 2400 source measurement unit (Tektronix^TM^, Beaverton, OR, USA). The applied drain and gate voltages were kept in the range of –1 V to +1 V to avoid electrolysis of water. Two types of measurements were performed: (i) source–drain current measurements during gate voltage sweeps between −1 V and +1 V in 0.1 V increments at constant drain voltage of +0.8 V, and (ii) source–drain current measurements at constant gate and drain voltages of +0.8 V. Both were optimized for maximum signal and stability in water and had an acquisition period time of approximately 0.2 s.

In the following, we refer to I_on_ as the current level at the minimum voltage in air (−60 V) or water (−1 V) and to I_off_ as the current level at the maximum voltage (+60 V in air, +1 V in water). Due to capacitive charging during the sweep, there may be two values of Ioff at +60 V (in air) or +1 V (in water); in the following, only the values obtained during the first upward sweep were used.

### 2.8. Measurements in Aqueous Solutions

The devices were tested in either 0.1 M of phosphate buffer (from pH 3 to pH 9) or 0.1 M borate buffer solutions (from pH 5 to pH 10) to assess how the pH response of CNTFETs depends on the buffer solutions. In the rest, deionized water refers to deionized water at 18 MW.cm at 25 °C.

Phosphate buffer solutions were prepared as follows: in 900 mL of deionized water (18 MΩ.cm at 25 °C), NaH_2_PO_4_·H_2_O and NaH_2_PO_4_ were added in amounts depending on the desired pH value (detailed compositions are given in Appendix A). NaOH (for increasing pH) or HCl (for decreasing pH) solutions (concentrations ~1M) were then added while continuously monitoring the pH until the solution reached the targeted pH value. Deionized water was then added to reach a total volume of 1 L (which had a negligible effect on the pH due to the high molarity of the buffer).

Borate buffer solutions were prepared as follows: 6.2 g of Boric acid (molecular weight: 62 g/mol) was added to 900 mL of deionized water (18 MΩ.cm at 25 °C). Then, NaOH (for increasing pH) (concentration ~1 M) was added in increasing amounts until the solution reached the desired pH value. Deionized water was then added to reach a total volume of 1 L.

For measurements, the devices were placed in a beaker and connected to the electronic acquisition system. Magnetic stirring was not used to avoid electrical interference. The pH was continuously recorded using a computer-connected pH meter ENV-40-pH, AtlasScientific^TM^, Long Island City, NY, USA) The pH meter was calibrated with commercial buffer reference solutions at the beginning of each day of measurements. For each pH step, the number of current measurement points at fixed gate and drain voltages was set at 3000. Each step lasted approximately 600 s. This duration was chosen to allow stabilization of the current level for each pH value.

## 3. Results

### 3.1. Electrical Characterization of CNTFETs in Air (Ambient Conditions)

The measured I_on_/I_off_ ratio ranges from 1 to 10 for p-CNTFET and from 50 to 500 for f-CNTFET. The passivation clearly improves the I_off_ value, leading to an increase in the I_on_/I_off_ ratio, as expected from the literature [43]: up to two orders of magnitude of increase for p-CNTFET and one order for f-CNTFET (Figure 4). More details on the I–V transfer curves (effect of passivation on the transfer curves, I_on_ and I_off_ as a function of channel length and of passivation) are given in Appendix A.

As often reported [46,47,48] for handmade devices, and as observed in Figure 5, I_on_ and I_off_ values are found to be variable from device to device for both types of FETs. However, normalized transfer curves (current level divided by I_on_) are very repeatable, as can be observed in Figure 5 showing I/I_on_ as a function of V_gs_.

Repeatability is quantified by calculating the average hysteresis of the transfer curves per device type and then by calculating the standard deviation over this average hysteresis value. Hysteresis is defined here as the voltage difference between the up and down sweeps at a current equal to I_on_/2 (e.g., full width at half maximum). The results are expressed in Table 2: while the hysteresis is larger for f-CNTFET, the repeatability is better. The repeatability is also better for smaller channel length. For all device types, the hysteresis is large compared to the measurement range (−60 V to +60 V). It is attributed to the bottom-gated structure, as the CNT/insulator interface is exposed to moisture and surface defects are not passivated, resulting in strong charging effects [49]. In the present structure, the SiO_2_/Si_3_N_4_ interface may also contribute to the hysteresis by charging the interface defects [50,51].

The subthreshold slope is similar for p-CNTFET and f-CNTFET, 80 V/decade and 82 V/decade, respectively, when calculated on the downward slope (from −60 V to +60 V). The effect of passivation on hysteresis and subthreshold slope has been quantified for 5 µm devices: the hysteresis decreases by 50% for p-CNTFETs and 15% for f-CNTFETs, and the subthreshold slope increases up to 15% for both p-CNTFETs and f-CNTFETs.

### 3.2. Electrical Characterization of CNTFETs in Water

For characterization in water, only 5 µm-channel devices were investigated. This channel length was chosen instead of 10 µm because, in air, they had better device-to-device reproducibility and there was a similar performance between p-CNTFET and f-CNTFET (I_on_/I_off_ = 40–100 for p-CNTFET, 80–300 for f-CNTFET). As mentioned above, the voltage levels were systematically kept between ±1 V to avoid water electrolysis.

Figure 6 shows the I–V transfer curves for 5 μm- channel p-CNTFETs and f-CNTFETs at V_ds_ = + 0.8 V in pH 7 phosphate buffer solution (PBS), in logarithmic scale for the vertical axis (linear scale plot provided in Appendix A). Appendix A also shows the comparison of current levels in water at different gate voltages for a device without CNTs and a device with CNTs. It shows that the transistor effect in water is due to the CNT layer.

The pristine CNTFETs exhibited conventional p-type semiconductor transistor behavior with I_on_ = 1.4 × 10^−6^ ±·10^−7^ A and I_off_ = 1.6 × 10^−10^ ± 8 × 10^−11^ A (Figure 6 left), corresponding to an I_on_/I_off_ ratio of ~10^4^. They had a threshold voltage (V_th_) at ~−0.5 ± 0.04 V and a subthreshold slope of ΔV_gs_ ~100 mV/decade. In addition, the device-to-device repeatability was satisfactory (standard deviation of 20%, 50%, 20% and 5% for I_on_, I_off_, V_th_ and subthreshold slope, respectively). In comparison, in air, these devices had an I_on_/I_off_ ratio of ~10^2^ and a subthreshold slope of ~80 V per decade. This dramatic increase in performance is expected when switching from bottom-gated to electrolyte-gated FET [42,52].

On the other hand, f-CNTFETs have I_on_ = 1.6 × 10^−7^ ± 8 × 10^−8^ A, I_off_ = 2.3 × 10^−9^ ± 9 × 10^−10^ A, I_on_/I_off_ ~ 70, V_th_ at ~ −0.65 ± 0.08 V and a subthreshold slope of ΔV_gs_ ~ 200 ± 50 mV per decade (Figure 6 right). They had greater device-to-device variability, with one obvious outlier device in particular (standard deviation of 50%, 40%, 12% and 25% for I_on_, I_off_, V_th,_ and subthreshold slope, respectively). Remarkably, the f-CNTFETs devices immersed in buffer solutions showed ambipolar behavior, which was not observed in air or in p-CNTFET, with Dirac point (voltage at minimum current) V_Dirac_ = −0.1 ± 0.3 V and current at the Dirac point I_Dirac_ = 7.4 × 10^−10^ ± 1 × 10^−9^ A. The average value of I_on_/I_Dirac_ is ~210, three times larger than I_on_/I_off_ (~70). This feature is clear evidence of the proper functionalization of the SWCNT by FF-UR (except for possibly one device). This property was stable in PBS at pH 7 for at least three months, as detailed in Appendix D
Figure A3 and Figure A4. The mechanisms involved in this ambipolar response are discussed in Section 3.4.3 after comparison of the I–V curves in different buffer solutions.

### 3.3. pH Sensitivity in PBS and BBS

#### 3.3.1. Reversible Response in PBS

The pH response was tested for both p-CNTFET and f-CNTFET in PBS with a pH ranging from pH 3 to pH 9 and then from pH 9 to pH 3. Figure 7 shows the real-time-measured pH response for 5 μm channel p-CNTFET. We observe that the measured current stabilizes after hundreds of seconds. The current level for each pH step was calculated as the average of the last 10% of each step (Appendix A). The relative sensitivity was calculated as the slope of the curve (I versus pH) divided by the current at pH 7. Note that the long stabilization time may be due to two factors, one being the reorganization of the electrical double layer as the electrolyte composition changes, and the other being acid-base reactions along the CNT surface with variable reaction rates.

Figure 8 shows normalized current values of p-CNTFET and f-CNTFET between pH 3 and pH 9 with linear fitting lines. The pristine CNTFET showed a linear, reversible pH response between pH 3 and pH 9 (Figure 8 left and Table 3). The relative sensitivity is 26 ± 2.2%/pH unit for upward pH and 21 ± 2.1%/pH unit for downward pH with an average relative sensitivity of 23.5 ± 2.2%/pH unit. When the pH was cycled between pH 6 and pH 9 over 3 days and five cycles, good reversibility of the four p-CNTFET devices was observed after a settling phase over the first cycle (Appendix E Figure A5).

In contrast, the f-CNTFETs had a much stronger, reversible pH response (370%), but only over the range from pH 7 to pH 9. The response from pH 3 to pH 7 is much less sensitive and reversible than that of p-CNTFET (Figure 8 right and Table 3). It is worth mentioning that the choice of pH 7 as the current reference enhances the relative sensitivity of f-CNTFET compared to p-CNTFET because the sensitivity range of f-CNTFET only starts from pH 7 upward.

#### 3.3.2. Response in BBS

For comparison, the sensors were also tested in borate buffer solution (BBS) from pH 5 to pH 10. Both p-CNTFET and f-CNTFET showed a linear pH response in BBS, with a sensitivity of 56%/pH for p-CNTFET and 96%/pH for f-CNTFET (Figure 9 and Table 4). This result shows that the composition of the pH buffer affected the pH sensing capability. The performance of the f-CNTFET remained significantly better than that of p-CNTFET (+58% relative difference), now with good sensitivity over the entire targeted pH range.

### 3.4. I–V Curves as a Function of pH in PBS and BBS

#### 3.4.1. p-CNTFET

To understand the sensing mechanism in PBS and BBS, the I–V transfer curves were measured at different pH values. 

Figure 10 and Figure 11 show the results in log-lin scale, while the results in lin-lin scale are given in Appendix A. Figure 12 compares the V_th_ and subthreshold slopes in the different configurations. The threshold voltage was defined as the gate voltage to reach I_on_/10 (one decade current reduction).

Focusing first on p-CNTFETs, the main effect of pH in both PBS and BBS is to increase the threshold voltage (~ +10 mV/pH unit and ~ +40 mV/pH unit, respectively), with slight variations in the subthreshold slope (between 100 and 150 mV/decade). This increase in the threshold voltage with pH explains the pH sensitivity of the current described in the previous sections. The higher threshold voltage in BBS explains the higher current sensitivity than in PBS.

These results for pristine electrolyte-gated CNTFET are fully consistent with the literature, as reported and modeled in [20,21]. This is attributed to the increase in charge density on the CNT outer surface due to the deprotonation of carboxylate groups (pKa ~ 4.5) with increasing pH, which in turn leads to increased p-doping. The higher threshold voltage in BBS is due to the lower ionic strength compared to the phosphate buffer. The range of the subthreshold slope, between 100 and 150 mV/decade, is higher than the theoretical limit of 60 mV/decade, indicating a moderate gating efficiency between 0.4 and 0.6.

#### 3.4.2. Comparison Between f-CNTFET and p-CNTFET

The comparison between p-CNTFET and f-CNTFET in BBS shows similar threshold voltages except at pH 5, where the threshold voltage is much lower than for p-CNTFET. The subthreshold slopes are in very good agreement above pH 7, but very different below: it decreases linearly from pH 5 to 7 starting from a high 300 mV/decade, indicating a low gating efficiency (0.2). These trends explain the increased current sensitivity to pH of f-CNTFETs compared to p-CNTFETs in BBS. The large decrease in subthreshold slope is consistent with the large increase in threshold voltage, as both are consistent with increased negative charge on the CNT surface with increasing pH, which in turn increases effective p-doping and gating efficiency. However, there is a need to explain why this effect is enhanced in f-CNTFETs compared to p-CNTFETs, which will be addressed below.

The difference between p-CNTFET and f-CNTFET is even more pronounced in PBS: n-type conduction appears at the three pH levels, while the intensity of the n-type ON current increases with pH. The n-type I_on_/I_off_ ratio remains very small compared to p-type, so the n-type threshold voltage cannot be estimated except at pH 7 (0.1 V), but the shift voltage (transition voltage between p-type conduction and n-type conduction, estimated as the voltage for minimum current) can be estimated: −0.1 V at pH 4 and 7, and +0.4 V at pH 10. Furthermore, at pH 7, the gap width is ~1 V. Meanwhile, the p-type threshold voltage increases with pH (faster than for p-CNTFET); the p-type subthreshold slope also strongly increases with pH: at pH 4, it is close to the theoretical limit; at pH 7, it is ~100 mV/decade, similar to f-CNTFET in BBS and to p-CNTFET in both buffers; at pH 10, it reaches 400 mV/decade, indicating a very low gating efficiency.

#### 3.4.3. Role of FF-UR in the pH Response of f-CNTFET

The appearance of n-type conduction in the f-CNTFETs in PBS is characteristic of the reduction of the effective bandgap of the functionalized SWCNT layer. Considering a gating efficiency of ~0.6 at pH 7 and a gap width of ~1 V, the bandgap can be estimated to be ~0.6 eV. Considering the differences between f-CNTFET responses in PBS and in BBS, this cannot be directly attributed to the FF-UR polymer alone, but to the coupling of FF-UR with an interferent in the water matrix. This effect is most likely due to a strong interaction of FF-UR with H_2_PO_4_^−^ and HPO_4_^2−^. Indeed, this interaction was found to be numerically very favorable in water by Benda et al. [40]. The modification of the band gap would be caused by the fact that the highest unoccupied orbitals of H_2_PO_4_^−^ and HPO_4_^2−^ fall within the band gap of the (semiconducting) SWCNTs [52].

Furthermore, the ratio of H_2_PO_4_^−^ and HPO_4_^2−^ ions complexed by FF-UR is expected to strongly depend on the relative concentration between the two species, which is highly dependent on whether the pH is higher or lower than the pKa of H_2_PO_4_^−^/HPO_4_^2−^ (~7). The type of phosphate ion complexing FF-UR directly affects the surface charges of the CNT, which control the threshold voltage and the level of p-doping: H_2_PO_4_^−^ below pH 7 has a single charge, like the carboxylate moieties responsible for the pH-sensitivity of pristine-CNTFETs. On the other hand, HPO_4_^2−^ above pH 7 has two negative charges and thus enhances the p-doping within the CNT; however, the more intense surface charge results in stronger electrostatic screening effect, leading to the degraded subthreshold slope at higher pH. This is in perfect agreement with the very different current trends observed below and above pH 7.

In the case of BBS, there is no evidence of a change in the band structure of FF-UR-functionalized SWCNTs due to the presence of boric acid (which is the majority form of boric acid below pH 9; the concentration of dihydrogen bromide ions increases with pH starting from pH 7 upward). However, the strong change in the subthreshold slope between pH 5 and 7—indicative of a screening effect—and the much lower threshold voltage at pH 5—indicative of reduced p-doping—compared to the pristine CNT case, suggest some affinity of FF-UR with bromic acid (H_3_BO_3_). At lower pH, bromic acid, being chargeless, would provide a low surface charge. Its presence would also increase the distance between the first solvated ion layer and the CNT surface, resulting in reduced gating efficiency. However, above pH 7, the fact that the threshold voltage and subthreshold slope of f-CNTFET in BBS are extremely close to those of p-CNTFET in BBS and PBS, and f-CNTFET in BBS, suggests a low affinity between FF-UR and dihydrogen bromide ions. Note that the interaction between FF-UR and the different forms of bromic acid was not investigated by Benda et al. [40].

To summarize, the study of the I–V curves as a function of pH shows typical electrolyte-gated behavior for p-CNTFET, including buffer dependencies. However, the functionalization by FF-UR has a dramatic effect on the performance: the complexation by FF-UR of some the molecules present in the buffer modulates the p-doping and charge screening:Complexation of chargeless bromic acid lowers p-doping and reduces electrolyte gating efficiency;Single-charged dihydrogen phosphate ions provide similar screening and p-doping as the carboxylate moieties of SWCNTs;Doubly charged hydrogen phosphate ions provide intense screening and intense p-doping and reduce the effective band-gap of the SWCNTs.

## 4. Comparison to the State-of-the-Art and Discussion

Table 1 shows a quantitative comparison with the state-of-the-art. In PBS (the most commonly used pH buffer), the relative sensitivity of our p-CNTFET (effectively electrolyte-gated) is in perfect agreement with the best FET result reported in the literature: 23%/pH unit for a non-functionalized dual-gated CNTFET [22]. This indicates that the design, fabrication and characterization choices presented in this paper allow us to achieve a state-of-the-art performance even without functionalization. The different responses in PBS and BBS are also consistent with the literature, showing a strong dependence on the ionic strength of the solution.

Functionalization with FF-UR greatly improves the performance in both buffers: 96%/pH unit in BBS over the full pH range; and 370%/pH unit in PBS only from pH 7 to 9. This makes the f-CNTFETs presented here by far the most sensitive to date. While the FF-UR probe has no direct pH sensitivity, the strong sensitivity of FF-UR to its chemical environment causes a strong modification of the local environment of the CNT, which affects the efficiency of electrolyte gating and of the CNT doping. However, this mechanism of sensing raises the question of the pH sensing performance of such CNT-based devices in a real water matrix, where they would face a much wider range of charged and uncharged compounds.

## 5. Conclusions

A bottom-gated CNTFET design with embedded poly-Si gate and double SiO_2_/Si_3_N_4_ dielectric layer was proposed. Unsorted SWCNTs, either pristine or functionalized with a polyfluorene polymer bearing urea-based moieties, were inkjet-printed on the substrate to form the targeted percolating layer. The CNTFETs were passivated with a porous PMMA layer fabricated by the NIPS process.

In air, the unpassivated p-CNTFET and f-CNTFET exhibited p-type behavior over the gate voltage range between −60 V and +60 V. Their I_on_/I_off_ ratio ranged from 1 to 10 for p-CNTFET and from 50 to 500 for f-CNTFET. The passivation improved the I_off_ value, resulting in an increase in the I_on_/I_off_ ratio of up to two orders of magnitude for p-CNTFET and one order of magnitude for f-CNTFET.

In water, the operating range of the gate and drain voltages was severely limited to ±1 V due to water splitting but effective electrolyte gating allowed proper operation and modulation of the CNTFETs. The p-CNTFETs maintained a similar p-type behavior in PBS and BBS, while the f-CNTFETs exhibited p-type behavior in BBS and ambipolarity in PBS. Compared to air, in PBS, p-CNTFETs showed an increased I_on_/I_off_ ratio (~10^4^), while the f-CNTFETs showed a decreased I_on_/I_off_ ratio (~10^2^), although the I_on_/I_D_ ratio of ~10^3^ at the Dirac point (~−0.1 V) was comparable to the I_on_/I_off_ ratio in air. The ambipolarity of f-CNTFETs in PBS gradually decreased after several months of exposure time in deionized water (they became p-type), suggesting a slow degradation of the functional polymer in water.

The sensitivity to pH was evaluated by measuring the drain current at a gate voltage of −0.8 V and a drain voltage of −0.8 V. In PBS, p-CNTFETs showed a linear, reversible pH response between pH 3 and pH 9 with a sensitivity of 26 ± 2.2%/pH unit, while f-CNTFETs showed a much stronger, reversible pH response (370%/pH unit), but only over the range of pH 7 to pH 9. In BBS, both p-CNTFET and f-CNTFET showed a linear pH response between pH 5 and 9, with sensitivities of 56%/pH and 96%/pH, respectively.

Analyzing the I–V curve in BBS and PBS, the increased sensitivity to pH with FF-UR functionalization can be attributed to the interactions of FF-UR with hydrogen phosphate and dihydrogen phosphate ions in PBS and boric acid in BBS (the relative concentration of complexed species depends on pH). Complexation affects the efficiency of electrolyte gating and the surface charge around the CNT, both of which modify the I–V response of the CNTFET, resulting in the current sensitivity as a function of pH.

Overall, the performance of p-CNTFETs in PBS is comparable with the best results in the literature, while the performance of f-CNTFETs far exceeds the current state-of-the-art by a factor of nearly four in BBS and by more than a decade over a limited range of pH in PBS. This is the first time in the literature that a functionalization other than the intrinsic CNT carboxylate moieties has provided such state-of-the-art pH sensing performances in CNTFETs or CNT chemistors. On the other hand, this study also highlights the challenge of transferring this performance to a real water matrix, where many different species may compete for interactions with FF-UR. In such a case, extensive calibration procedures in the target matrix will be required.

## Figures and Tables

**Figure 1 nanomaterials-13-01157-f001:**
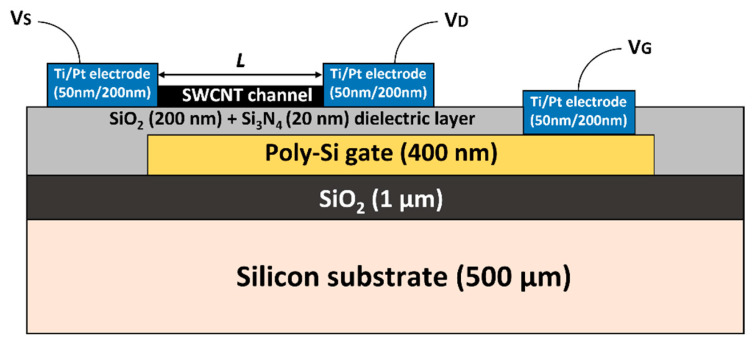
Schematic representation of the bottom-gated CNTFETdevices. Vs: Source voltage, V_D_: Drain voltage, V_G_: Gate voltage, L: Channel length varying from 5 µm to 120 µm.

**Figure 2 nanomaterials-13-01157-f002:**
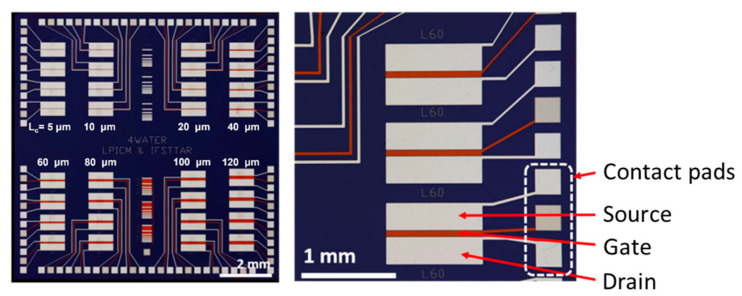
Pre-fabricated chip (10 mm by 10 mm) with gate, source and drain electrodes and contact pads for wire-bonding. Channel length L_c_ varies from 5 μm to 120 μm.

**Figure 3 nanomaterials-13-01157-f003:**
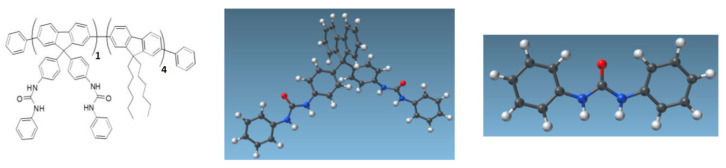
(**Left**): Chemical structure of FF-UR polymer. Adapted from [39], with the permission of AIP Publishing. (**Middle**): Fluorene monomer carrying two urea-based sensing moieties (SAMSON software). Reproduced from [40], with the permission of John Wiley and Sons. (**Right**): focus on the sensing moieties (SAMSON software). Atom color code: black for carbon, white for hydrogen, blue for nitrogen, red for oxygen. Reproduced from [40], with the permission of John Wiley and Sons.

**Figure 4 nanomaterials-13-01157-f004:**
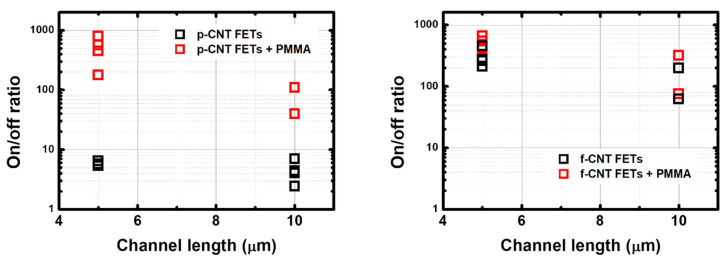
I_on_/I_off_ ratio of (**left**) p-CNTFETs and (**right**) f-CNTFETs before and after PMMA passivation with channel length of 5 μm and 10 μm.

**Figure 5 nanomaterials-13-01157-f005:**
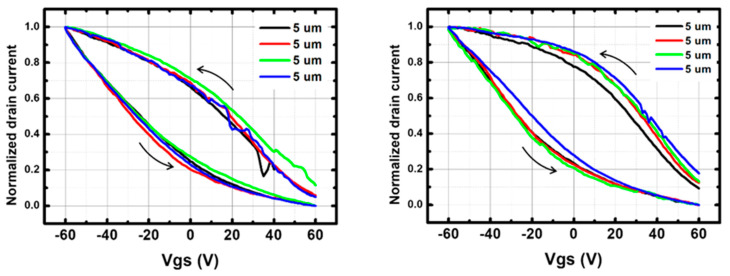
Normalized I–V transfer curves (current level divided by I_on_) of 5 µm channel (**left**) p-CNTFETs and (**right**) f-CNTFETs at drain–source voltage (V_ds_) of +5 V in air. Four different 5 μm transistors are shown for each type of FETs. The arrows indicate gate–source voltage (V_gs_) sweeping direction.

**Figure 6 nanomaterials-13-01157-f006:**
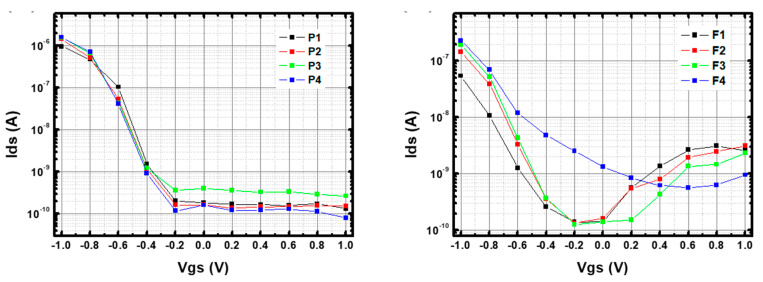
I–V transfer curve of (**left**) p-CNTFETs and (**right**) f-CNTFETs in phosphate buffer solution (PBS) at pH 7. All transistors were 5 μm channel devices. V_ds_ is set to +0.8 V and V_gs_ was swept from −1 V to +1 V. A logarithmic scale is used for the vertical axis.

**Figure 7 nanomaterials-13-01157-f007:**
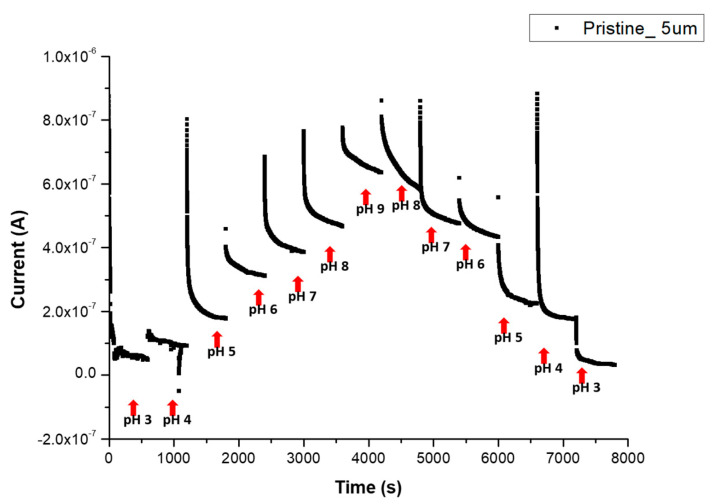
Real-time pH response measurement of p-CNTFET from pH 3 to pH 9 (upward direction) and from pH 9 to pH 3 (downward direction) in PBS. V_ds_ is set to +0.8 V and V_gs_ is set to −0.8 V.

**Figure 8 nanomaterials-13-01157-f008:**
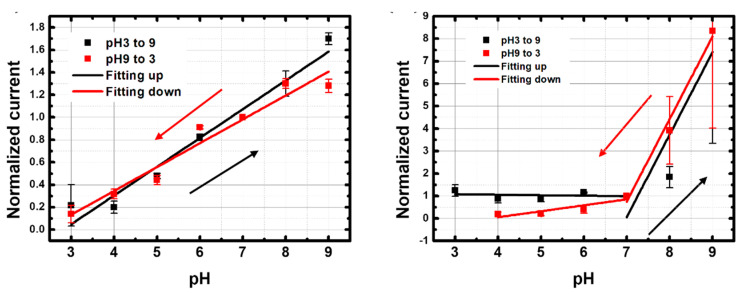
Normalized current values as a function of pH and linear fitting lines of (**left**) p-CNTFET and (**right**) f-CNTFET in phosphate buffer solutions (PBS) with increasing and decreasing pH from pH 3 to pH 9.

**Figure 9 nanomaterials-13-01157-f009:**
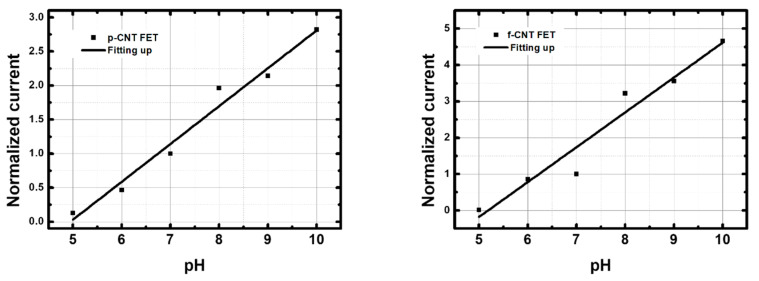
Normalized current values as a function of pH and linear fitting lines of (**left**) p-CNTFET and (**right**) f-CNTFET in borate buffer solution (BBS). Only a single direction from pH 5 to pH 10 was measured.

**Figure 10 nanomaterials-13-01157-f010:**
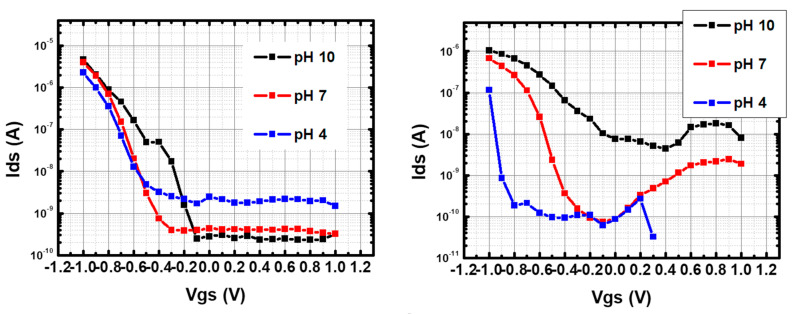
I–V transfer curve of (**left**) p-CNTFETs and (**right**) f-CNTFETs in PBS with respect to different pH from pH 10 to pH 4. V_ds_ is fixed at +0.8 V.

**Figure 11 nanomaterials-13-01157-f011:**
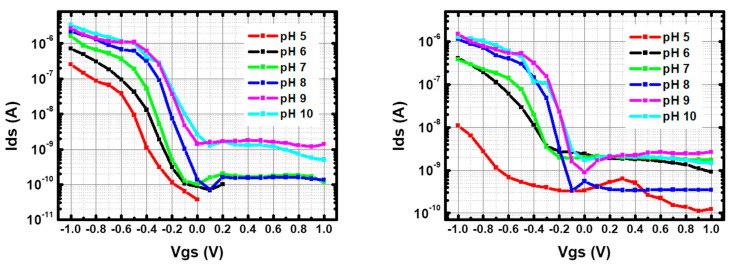
I–V transfer curve of (**left**) p-CNTFETs and (**right**) f-CNTFETs in BBS with respect to different pH from pH 10 to pH 4. V_ds_ is fixed at +0.8 V.

**Figure 12 nanomaterials-13-01157-f012:**
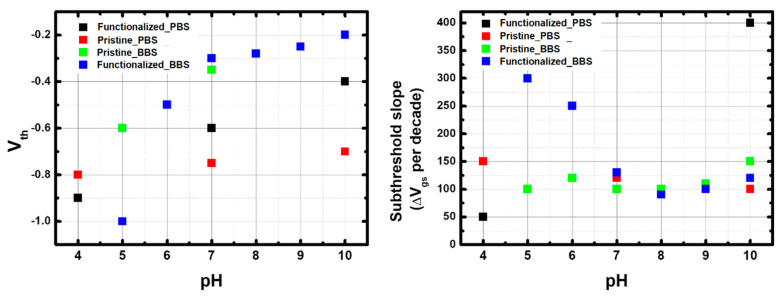
(**left**) V_th_ and (**right**) Subthreshold slope of p-CNTFETs and f-CNTFETs in different buffer solutions.

**Table 2 nanomaterials-13-01157-t002:** Average and standard deviation of hysteresis of p-CNTFETs and f-CNTFETs. Only 5 µm and 10 µm channel devices are considered.

Type of CNTFET	Channel Length			
	5 µm		10 µm	
	Average	Standard deviation	Average	Standard deviation
p-CNTFET	45 V	4.0 V (8.9%)	27 V	16 V (57%)
f-CNTFET	57 V	3.1 V (5.4%)	63 V	6.4 V (10%)

**Table 3 nanomaterials-13-01157-t003:** Calculated relative sensitivity of p-CNTFET and f-CNTFET from pH 3 to pH 9. The sensitivity of f-CNTFET is calculated in two different ranges, from pH 3 to pH 7 and from pH 7 to pH 9.

p-CNTFET			f-CNTFET		
Direction	Sensitivity (%/pH)	R^2^	Direction	Sensitivity (%/pH)	R^2^
Up (pH 3 → pH 9)	26 ± 2.2	0.96	Up (pH 3 → pH 7)	−2 ± 6	−0.28
			Up (pH 7 → pH 9)	370 ± 160	0.67
Down (pH 9 → pH 3)	21 ± 2.1	0.94	Down (pH 9 → pH 7)	370 ± 40	0.97
			Down (pH 7 → pH 3)	26 ± 9	0.71

**Table 4 nanomaterials-13-01157-t004:** Calculated sensitivity of p-CNTFET and f-CNTFET from pH 5 to pH 10 in borate buffer solution (BBS). Only upward direction was considered; 3.4. I–V curves as a function of pH in PBS and BBS.

p-CNTFET			f-CNTFET		
Direction	Sensitivity (%/pH)	R^2^	Direction	Sensitivity (%/pH)	R^2^
Up (pH 5 → pH 10)	56	0.97	Up (pH 5 → pH 10)	96	0.94

## Data Availability

The data presented in this study are available on request from the corresponding author.

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
