# Peer review of "Intense pH Sensitivity Modulation in Carbon Nanotube-Based Field-Effect Transistor by Non-Covalent Polyfluorene Functionalization"

_nanomaterials, 2023, doi:10.3390/nano13071157_

Round 1
Reviewer 1 Report
Manuscript ID nanomaterials-2279738
Report
The MS entitled “Intense pH sensitivity modulation in carbon nanotube-based field-effect transistor by non-covalent polyfluorene functionalization” by Gookbin Cho, Eva Grineval, Jean-Christophe P. Gabriel, and Bérengère Lebental reports on an experimental study of a new pH sensor. A new design of a CNTFET was proposed and demonstrated. The new CNTFET has a polysilicon bottom-gate CNTFET underneath a double dielectric layer (SiO2/Si3N4).
The paper is well organized, structured and written. An overview of similar technologies is given that allows to verify that the performance of the proposed device is better than the state of the art.
The results are relevant for researchers working in chemical sensors.
I recommend the paper for publication.
Author Response
Thank you for your comment
Reviewer 2 Report
This manuscript discusses a comprehensive and excellent review comparison of the pH sensing capabilities of non-functionalized carbon nanotube field-effect transistors (p-CNTFET) and CNTFET functionalized with a conjugated polyfluorene polymer (f-CNTFET). The devices were tested in different buffer solutions, and the results showed that f-CNTFET had a much stronger and reversible pH response, but only over a limited range of pH. The increased sensitivity of f-CNTFET was attributed to the interactions of the polymer with phosphate ions in PBS and boric acid in BBS. However, the study also highlighted the challenge of transferring these results to a real water matrix, where many different species may compete for interactions with the functionalized nanotubes. The results are important and useful. Before publication, I have few comment.
1. There are many channel material, such as Si, IZGO, 2D material for FET.
Why do author use CNTFET and has any special reason?
2. In what situation, we need such delicate device to detect pH value, comparing to conventional pH meter.
3. What is most import parameter (Ion, Ion/Ioff ration, voltage, voltage…..) to detect pH value, please emphasis on abstract and conclusion.
Author Response
Thank you for your review. Please find below the answer to your comments.
- There are many channel material, such as Si, IZGO, 2D material for FET. Why do author use CNTFET and has any special reason?
The interest of using CNT in chemical sensing applications, and more specifically for pH sensing, compared to other materials, is “due to their excellent mechanical and chemical stability, their large surface area and their chemical tunability, which enables selective sensing [6–8].”, as is stated in the introduction.
For extensive comparison to other types of FET, an additional reference has been added (ref 11: A comprehensive review of FET‐based pH sensors: materials, fabrication technologies, and modeling; Sinha S, Pal TElectrochemical Science Advances (2022) 2(5))
- In what situation, we need such delicate device to detect pH value, comparing to conventional pH meter.
Current commercial pH sensors are fragile (eg. use of porous glass for ionic exchange), require frequent maintenance (due to rapid evolution in time of the electrolyte materials), and expensive to produce (assembly of heterogenous component). As a consequence, the use of conventional pH sensors for online monitoring of drink water, while possible, is severely limited compared to current end-user needs.
Solid state devices such as FET or chemistor are actually expected to be less fragile and more stable than conventional pH electrodes. They would be suitable across the field of in-situ drink water monitoring solutions.
The introduction has been slightly modified to better carry this message
“Being solid-state devices (with no required storage of electrolyte), such electrical sensors are promising for future in-situ water monitoring solution: compared to commercial pH sensors, they are expected to be cheaper to fabricate and more easily stored, to require less calibration, to be more durable in field conditions, and to be less dependent upon the composition of the target water matrix. “
- What is most import parameter (Ion, Ion/Ioff ration, voltage, voltage…..) to detect pH value, please emphasis on abstract and conclusion.
We are measuring the drain current at a fixed drain and gate voltage values; both being higher than minimum voltage. This is stated in the abstract (“The sensitivity to pH is assessed by measuring the drain current at gate and drain voltage of -0.8 V.”) and the conclusion (“The sensitivity to pH was assessed by measuring the drain current at gate voltage -0.8 V and drain voltage -0.8 V.”).
Reviewer 3 Report
This paper compare the pH sensing performances of non-functionalized carbon nanotubes (CNT) field-effect transistors (p-CNTFET) and CNTFET functionalized with a conjugated polyfluorene polymer (labelled FF-UR) carrying urea-based moieties (f-CNTFET). The devices are electrolyte-gated, PMMA-passivated, 5 µm-channel FETs with unsorted, inkjet-printed single-walled CNT.
comment
1. Compared to other devices, a part with superior relative sensitivity is observed. However, as it shows a relatively low current characteristic, it is considered that the effect on noise is large. In addition, the pH measurement range is relatively narrow. It would be nice to write information about the pH 2-10 area as well.
2. When measuring air and water, it is necessary to explain the difference in the effect of capacitance according to the gate bias. Here, 60V and 1V were used as standards. Please explain whether the field effect of the two voltages is the same.
3. In the case of f-CNTFET, it is necessary to explain why it exhibits ambipolar characteristics.
4. Why is relaxation observed in the real time pH reaction, and it is necessary to compare the level in terms of reaction rate.
Author Response
Thank you for your comments.
- Compared to other devices, a part with superior relative sensitivity is observed. However, as it shows a relatively low current characteristic, it is considered that the effect on noise is large. In addition, the pH measurement range is relatively narrow. It would be nice to write information about the pH 2-10 area as well.
As is mentioned in the introduction, the target application is water quality monitoring, where the pH of interest is typically 5 to 9. The range 2 to 10 is beyond the application scope. Still the upper pH range was covered with borate buffer solutions from pH 5 to 10. On the other hand, the lower pH range was tested from pH 3 to 9 with phosphate buffer. PH 2 was not evaluated.
For the upper range of pH (7 to 10), the pH responses in borate buffer show a good linearity from 5 to 10 for p-CNTFET and f-CNTFET.
For the lower range of pH (3 to 7), pH responses in phosphate buffer show good linearity for p-CNTFET and no sensitivity for f-CNTFET.
The quite high standard deviation of the relative sensitivity of f-CNTFET in phosphate buffer reflects the fact that for this type of devices, in PBS, the range of study is limited from pH 7 to 9 and that the standard deviation on each point significant.
- When measuring air and water, it is necessary to explain the difference in the effect of capacitance according to the gate bias. Here, 60V and 1V were used as standards. Please explain whether the field effect of the two voltages is the same.
This is the difference between a bottom gated and a liquid gated structure, as is detailed in ref 20 and 21. In the former, the capacitance is controlled by the (solid) dielectric layer thickness and dielectric constant. In the latter, the capacitance is controlled by the electrical double layer thickness and dielectric constant. The text has been slightly modified to emphasize this aspect.
“Most CNTFET used as pH sensors are liquid-gated [20], as this approach has the simplest fabrication route. It also provides outstanding transistor performances, as the electrical double layer forming in the electrolyte acts as dielectric layer [21].”
The current level Ion in air (-60V) is higher than in water (-1V) by a factor of about 10 in the different devices. However, the difference cannot be directly attributed to a different field effect, as in an electrolyte-gated CNTFET, the density of charge close to the surface (which depends on the exact electrolyte and on the gate voltage) also impacts the density of charge
- In the case of f-CNTFET, it is necessary to explain why it exhibits ambipolar characteristics.
This is detailed in section 3.4.3
The apparition of n-type conduction in f-CNTFET in PBS is characteristic of reduction of the effective bandgap of the functionalized SWCNT layer. Considering a gating efficiency of ~0.6 at pH 7 and a gap width of ~1 V, the bandgap can be estimated at ~0.6 eV. Considering the differences between f-CNTFET responses in PBS and in BBS, this cannot be directly linked to the FF-UR polymer on its own, but to the coupling of FF-UR with an interferent in the water matrix. This effect is most likely due to a strong interaction of FF-UR with H2PO4- and HPO42-. This interaction was indeed found numerically to be very favorable in water by Benda et al. [41]. The modification of the bandgap would be caused by the highest unoccupied orbitals of H2PO4- and HPO42- falling within the bandgap of the (semi-conducting) SWCNT [55].
- Why is relaxation observed in the real time pH reaction, and it is necessary to compare the level in terms of reaction rate.
In electrolyte-gated FET, the real-time pH response is due, in part, to the reorganization of the electrical double layer when the electrolyte composition changes. While there may be some contribution from the acid-base reaction kinetics, the two effects may not be separated in this experimental design. Studying the reaction rates of this system would require development of completely different protocols, for instance AC operation of the FET (similar to what is done in impedance spectroscopy), which goes beyond the scope of the present paper.
A comment on that has been added in section 3.3.1
Note that the long stabilization time may have two contributing factors, on the one hand the reorganization of the electrical double layer when the electrolyte composition changes, on the other hand acid-base reactions along the CNT surface with variable reaction rates.